# Antimicrobial Activity of N,N-Diethyldithiocarbamate against *Ureaplasma parvum* and *Ureaplasma urealyticum*

**DOI:** 10.3390/ijms25010040

**Published:** 2023-12-19

**Authors:** Małgorzata Biernat-Sudolska, Danuta Rojek-Zakrzewska, Kamil Drożdż, Anna Bilska-Wilkosz

**Affiliations:** 1Department of Molecular Microbiology, Faculty of Medicine, Jagiellonian University Medical College, 31-121 Krakow, Poland; malgorzata.biernat-sudolska@uj.edu.pl (M.B.-S.); danuta.rojek-zakrzewska@uj.edu.pl (D.R.-Z.); kamil.drozdz@uj.edu.pl (K.D.); 2Chair of Medical Biochemistry, Faculty of Medicine, Jagiellonian University Medical College, 31-034 Krakow, Poland

**Keywords:** *Ureaplasma urealyticum*, *Ureaplasma parvum*, N,N-diethyldithiocarbamate, disulfiram

## Abstract

*Ureaplasma* species (*Ureaplasma* spp.) are commonly found as commensals in the human urogenital tracts, although their overgrowth can lead to infection in the urogenital tract and at distal sites. Furthermore, ureaplasmas lack a cell wall and do not synthesize folic acid, which causes all β-lactam and glycopeptide antibiotics, and sulfonamides and diaminopyrimidines, to be of no value. The antibiotics used in therapy belong to the fluoroquinolone, tetracycline, chloramphenicol and macrolide classes. However, the growing incidence of antibiotic-resistant *Ureaplasma* spp. in the population becomes a problem. Thus, there is a need to search for new drugs effective against these bacteria. Since 1951, the FDA-approved, well-tolerated, inexpensive, orally administered drug disulfiram (DSF) has been used in the treatment of chronic alcoholism, but recently, its antimicrobial effects have been demonstrated. The main biological metabolite of DSF, i.e., N,N-diethyldithiocarbamate (DDC), is generally believed to be responsible for most of the observed pharmacological effects of DSF. In the presented studies, the effect of DDC at concentrations of 2 µg/mL, 20 µg/mL and 200 µg/mL on the growth and survival of *Ureaplasma urealyticum* and *Ureaplasma parvum* was tested for the first time. The results indicated that all the used DDC concentrations showed both bacteriostatic and bactericidal activity against both tested strains.

## 1. Introduction 

*Ureaplasma* is a genus of bacteria belonging to the class Mollicutes. *Ureaplasma* species (*Ureaplasma* spp.) are commonly found as commensals in the human urogenital tracts, although their overgrowth can lead to infection in this system and at distal sites [1,2,3,4]. 

These microorganisms inhabit also the respiratory tract of newborns, and their presence may cause numerous life-threatening diseases [5]. There are two species that occur in humans, *Ureaplasma urealyticum* (*U. urealyticum*) and *Ureaplasma parvum* (*U. parvum*). Treatment of ureaplasma infections is complicated. This is due to the fact that ureaplasmas lack a cell wall and do not synthesize folic acid, which causes all β-lactam, glycopeptide antibiotics and sulfonamides and diaminopyrimidines, to be of no value. The antibiotics used in therapy include macrolides, fluoroquinolones and tetracyclines. However, many authors point to the increasing incidence of antibiotic-resistant *Ureaplasma* spp. infections in the population. Research by Khosropour et al. indicated that 57% of people with nongonococcal urethritis (NGU) infected with *Ureaplasma* spp. who received antibacterial therapy initially with azithromycin and then doxycycline for 7 days, or vice versa, were still colonized with these bacteria after 6 weeks of treatment [6]. Biernat-Sudolska et al. showed that as many as 90% of *U. parvum* strains and 83% of *U. urealyticum* strains present in the tracheal aspirate of premature newborns were resistant to ciprofloxacin belonging to the fluoroquinolone class [7]. Yang et al. indicated that the resistance rates of levofloxacin were 82.43% for *U. urealyticum* and 84.69% for *U. parvum*, and those of moxifloxacin were 62.16% for *U. urealyticum* and 51.44% for *U. parvum* [8]. There is, therefore, a need to search for new drugs effective against these atypical bacteria. The development of a new antimicrobial is a very time-consuming process. Efficacy studies and safety profile assessments may take up to even several years for one drug. Repurposing of already approved drugs for new medical uses is therefore a promising alternative strategy as it takes advantage of existing toxicology and pharmacokinetic data from preclinical and clinical trials. For this reasons, it is a faster and more economical approach. Disulfiram (DSF) seems to be an attractive drug in this context. DSF was approved in 1951 by the US Food and Drug Administration (FDA) as a drug for the treatment of alcoholism. It has been widely used in clinics for over 70 years as a well-tolerated drug, inexpensive, and without severe side effects. Recently, it has been shown that DSF and its metabolites can exert antimicrobial effects [9]. The main biological metabolite of DSF is N,N-diethyldithiocarbamate (DDC). One molecule of DSF gives rise to two molecules of DDC via the reduction of an intramolecular disulfide bond. In the blood of humans and other animals, DSF is rapidly and completely converted into two molecules of DDC by serum albumin and erythrocyte enzymes [10,11]. This means that the molecule of DSF practically does not exist in the body after ingestion and should be treated as a pharmacologically inactive prodrug that is metabolized in the body to pharmacologically active compounds [12]. 

So, DDC is the main metabolite of DSF and is generally believed to be responsible for most of the observed pharmacological effects of DSF. An additional advantage of DDC is the lack of teratogenic, mutagenic or carcinogenic effects in animal models [13]. DDC is the reduced form of DSF. The chemical structures of DSF and DDC are shown in Figure 1.

Thus, DDC seems to be a compound with great pharmacological potential. Considering the above, the aim of the present study was to determine for the first time whether DDC influences the growth and multiplication of *U. urealyticum* and *U. parvum* strains. 

## 2. Results 

### 2.1. The Bacteriostatic and Bactericidal Effect of DDC on Ureaplasma Cultures

#### 2.1.1. The Bacteriostatic Effect of DDC on Ureaplasma Cultures

The median value of the log titer for the control *U. parvum* cultures was 3 (2, 5) while the median values of the log titers for *U. parvum* cultured in the presence of DDC at concentrations of 200 μg/mL, 20 μg/mL, and 2 μg/mL were 1 (0, 2), 1 (0, 1) and 2 (1, 3), respectively (Figure 2). Statistically significant differences were observed at all the tested concentrations of DDC on *U. parvum* when compared to the control cultures without DDC (*p* < 0.05). Furthermore, the concentrations of 20 μg/mL (*p* = 0.001) and 200 μg/mL (*p* < 0.001) showed a statistically significantly greater bacteriostatic effect compared to the 2 μg/mL concentration. The median value of the log titer for the control *U. urealyticum* cultures was 3 (2, 6) while the median values of the log titers for *U. urealyticum* cultured in the presence of DDC at concentrations of 200 μg/mL, 20 μg/mL, and 2 μg/mL were 1 (0, 1), 1 (1, 2) and 2 (1, 5.75), respectively (Figure 2). Each concentration of DDC tested exhibited a statistically significant impact on *U. urealyticum*, as evidenced by the differences compared to the control cultures without DDC (*p* < 0.001), with the concentrations of 20 μg/mL (*p* < 0.001) and 200 μg/mL (*p* = 0.001) showing a statistically significantly greater effect on cell division compared to the 2 μg/mL concentration.

Additionally, statistical analyses revealed that there were no significant differences between *U. parvum* and *U. urealyticum* at the same concentrations of DDC. This finding underscores the similar effect of DDC on both strains of bacteria at each tested concentration.

The obtained results therefore showed that the presence of DDC has the same bacteriostatic effect (cell division) on both species tested, regardless of the concentration used.

#### 2.1.2. The Bactericidal Effect of DDC on Ureaplasma Cultures

The median value of the log titer for the control *U. parvum* cultures was 3 (1, 5) while the mean values of the log titers for *U. parvum* cultured in the presence of DDC at concentrations of 200 μg/mL, 20 μg/mL, and 2 μg/mL were 1 (0, 2), 0 (0, 2) and 1 (0, 3), respectively (Figure 2). Statistically significant differences were observed at all the tested concentrations of DDC on *U. parvum* compared to the control cultures without DDC (*p* < 0.05); however, the statistical analysis did not reveal any statistically significant variations among the concentrations tested.

The median value of the log titer for the control *U. urealyticum* cultures was 3 (1, 5.75) while the median values of the log titers for *U. urealyticum* cultured in the presence of DDC at concentrations of 200 μg/mL, 20 μg/mL, and 2 μg/mL were 0 (0, 1), 1 (0, 2) and 2 (0, 3.75), respectively (Figure 2). For *U. urealyticum*, statistically significant differences were observed at all the concentrations of DDC tested when compared to the control cultures without DDC. Furthermore, the concentrations of 20 μg/mL (*p* = 0.029) and 200 μg/mL (*p* = 0.001) demonstrated statistically significant differences compared to the 2 μg/mL concentration.

Moreover, statistical analysis showed that the bactericidal activity, expressed in log titer values, did not differ significantly against *U. parvum* and *U. urealyticum* under the influence of identical DDC concentrations. This suggests the comparable bactericidal effectiveness of the assessed DDC concentrations against both species of ureaplasmas. 

The obtained results therefore showed that the presence of DDC had a bactericidal effect on both tested species, regardless of the tested concentration.

### 2.2. The Percentage of U. parvum (Up) and U. urealyticum (Uu) Strains for Which Bacteriostatic and Bactericidal Effects of DDC Were Demonstrated

#### 2.2.1. The Percentage of *U. parvum* (Up) Strain for Which Bacteriostatic (Cell Division) and Bactericidal Effects of DDC Were Demonstrated

The results showed that all the DDC concentrations tested had both bacteriostatic and bactericidal effects against the species *U. parvum*. The bacteriostatic effect of DDC at doses of 2, 20 and 200 µg/mL was observed against 68.4%, 88.9% and 72.2% of strains belonging to this species, respectively. The strongest bacteriostatic effect of DDC was observed using 20 µg/mL. The results obtained using a concentration of 20 µg/mL were statistically significantly higher compared to the results obtained at a concentration of 2 µg/mL (*p* = 0.039), while no statistically significant differences in effectiveness were observed between the concentrations of 200 µg/mL and 20 µg/mL (*p* = 0.808).

The bactericidal effect of DDC at doses of 2, 20 and 200 µg/mL was observed against 42.0%, 54.0% and 46% of the tested strains, respectively. Although the strongest bactericidal effect against the *U. parvum* species was also observed at a dose of 20 µg/mL, this difference was not statistically significant compared to the concentrations of 2 µg/mL (*p* = 0.239) and 200 µg/mL (*p* = 0.430).

However, the analysis of the obtained results in relation to χ2 clearly showed that the bacteriostatic effect of DDC on the *U. parvum* strain is much stronger than the bactericidal effect (Figure 3) (*p* < 0.05).

#### 2.2.2. The Percentage of *U. urealyticum* (Uu) Strain for Which Bacteriostatic (Cell Division) and Bactericidal Effects of DDC Were Showed

The results showed that all the DDC concentrations tested had both bacteriostatic and bactericidal effects against the species *U. urealyticum* (Figure 4). The bacteriostatic effect of DDC at doses of 2, 20 and 200 µg/mL was observed against 61.1%, 97.2% and 88.9% of strains of this species, respectively. Therefore, the strongest bacteriostatic effect against *U. urealyticum* was obtained at a dose of 20 µg/mL, with a statistically significant difference compared to the concentration of 2 µg/mL (*p* = 0.030), while no statistically significant differences were observed between the concentrations of 20 µg/mL and 200 µg/mL (*p* = 0.173).

The bactericidal effect of DDC on *U. urealyticum* species at doses of 2, 20 and 200 µg/mL was observed for 28.1%, 46.9% and 59.4% of strains, respectively. Therefore, the most effective bactericidal effect against *U. urealyticum* was obtained at a dose of 200 µg/mL, and the difference between 200 µg/mL and 20 µg/mL was statistically significant (*p* = 0.031), but no statistical significance was found when comparing the concentration of 200 µg/mL up to 2 µg/mL (*p* = 0.324).

However, the analysis of the obtained results in relation to χ2 clearly showed that the bacteriostatic effect of DDC on the *U. urealyticum* strain is much stronger than the bactericidal effect (*p* < 0.05).

## 3. Discussion

The obtained results clearly demonstrated that all the tested DDC concentrations showed both bacteriostatic and bactericidal activity against both tested species.

The highest bacteriostatic effect against *U. urealyticum* (97.2%) and *U. parvum* (88.9%) was observed at a DDC concentration of 20 µg/mL. The differences in the response of the tested strains to the bacteriostatic effect of DDC are not statistically significant (*p* > 0.05). This means that the bacteriostatic effect of DDC on *U. parvum* and *U. urealyticum* cells is comparable. The strongest bactericidal effect against *U. urealyticum* (59.4%) was observed at a DDC concentration of 200 µg/mL. The highest bactericidal effect of *U. parvum* cells (54.0%) was observed at a DDC concentration of 20 µg/mL. This means that DDC has a stronger bactericidal effect on *U. urealyticum* cells than on *U. parvum* cells.

The data contained in the Section 4 present the bactericidal effect of DDC, illustrating the percentage of tested strains grown in the presence of DDC, which did not retain the ability to grow after being transferred to a culture medium without the addition of DDC.

The antimicrobial effect of DSF and its main metabolite DDC has been known for a long time. Already in 1974, it was established that DSF showed fungicidal activity [14]. In 1979, Scheibel et al. revealed that DSF inhibited in vitro growth of the human malaria parasite *Plasmodium falciparum* [15]. In 1987, Taylor et. al., demonstrated that DDC inhibited the growth of methicillin-resistant *Staphylococcus aureus* in vitro [16]. Kobatake et al. recently reported that DSF had a bactericidal effect on *Helicobacter pylori* [17]. 

Repurposing “old” drugs to treat “new” diseases is becoming an increasingly attractive strategy. In other words, drugs approved for the treatment of one ailment are used to treat others. Therefore, existing evidence suggests that DSF, which is an approved drug for the treatment of alcoholism, can be successfully used in the treatment of infections caused by various microorganisms [9]. Further attempts to use DSF and its metabolites in the treatment of bacterial infections are therefore justified.

So far, most studies on the antibacterial properties of DSF and its metabolites have focused primarily on typical Gram-positive and Gram-negative bacteria. Microorganisms of the *Ureaplasma* genus do not have a cell wall and therefore are not stained with the Gram method. We found only one study in various bibliographic databases showing that DSF revealed copper-dependent antimicrobial activity against *Mycoplasma hominis*, a species of bacteria belonging to the same class Mollicutes, which also includes the genus *Ureaplasma* [18]. Hence, our interest in this compound stems from the above data.

As already mentioned, the results of our studies indicate that all the DDC concentrations used in the presented experiments had both bacteriostatic and bactericidal effects against the tested clinical strains belonging to both human *Ureaplasma* spp. Our studies also showed that the bacteriostatic effect of DDC on the tested strains was stronger than the bactericidal effect. It is also worth noting that the strongest bacteriostatic effect was observed at a DCC dose of 20 µg/mL. Increasing the dose to 200 µg/mL did not increase its bacteriostatic effect, and this effect was even worse than at a dose of 20 µg/mL. The phenomenon of a lack of a linear dose–effect relationship is often observed in studies on the antimicrobial effects of antibiotics. The lack of a linear relationship between the dose used and the effect of the preparation means that with an increase in the dose of the drug, there is no increase in the effectiveness of the drug or, as in our studies, the paradoxically decreasing effectiveness of the drug is observed. This situation was also presented in the study by Dégrange et al., which showed that as the dose of tetracycline increased, its antimicrobial effectiveness against *Mycoplasma hominis* bacteria decreased [19]. Hillier et al. found a higher resistance rate associated with higher doses of amoxicillin [20]. 

The highest concentrations of DDC used had a bactericidal effect on over 50% of strains of the *U. urealyticum* cells. The *U. urealyticum* species, although less common, is considered more pathogenic, especially when it comes to infections in newborns, and is usually more drug-resistant [5,21,22,23]. Moreover, the analysis of the obtained results clearly showed that the bacteriostatic effect of DDC on the *U. urealyticum* and *U. parvum* strains is much stronger than the bactericidal effect. DDC at a dose of 20 μg/mL is bacteriostatic for 97.2% of *U. urealyticum* cells and 88.9% of *U. parvum* cells. Therefore, the obtained results are promising.

Although the antimicrobial effect of DSF and its metabolites is increasingly demonstrated by various authors, the mechanism of this action is not yet well understood. Two mechanisms are most often taken into account. 

One is related to the ability of DSF and its metabolites to react with the thiol groups of proteins present in the microbial cell [9]. DSF, tetraethylthiuram disulfide, is a disulfide, and DDC, diethyldithiocarbamate, is a reduced form of DSF, i.e., it is a thiol (Figure 1). Chemical compounds of this type are known to form mixed disulfides with protein thiols/disulfides. This irreversible reaction may modify catalytic cysteine residues on the target enzymes, causing the inhibition of their enzymatic activity. The existence of this type of mechanism has been demonstrated in many studies. Hamblin et al. showed that DSF was able to inhibit the activity of the aldehyde dehydrogenase (ALDH)-like protein in *Francisella tularensis* [24]. It has also been shown that betaine aldehyde dehydrogenase is a potential target for antimicrobial agents against *Pseudomonas aeruginosa*. The study demonstrated that DSF was an inhibitor of this enzyme in *Pseudomonas aeruginosa* cells [25]. 

The second mechanism considered to explain the antimicrobial action of DSF and its metabolites is that DDC, as a thiol compound, is able to chelate metal ions. Through the chelation of vital metals, DDC can, therefore, impair the function of metal-containing enzymes [26]. 

The proposed ureaplasma virulence factors include IgA, protease, urease, and phospholipases A and C [27]. Therefore, the plausible proposal is that IgA, protease, urease, and phospholipases A and C are also potential targets for antimicrobial agents, including DDC against *Ureaplasma* spp. 

The analysis of the results shown in Figure 2 indicates in several cases large deviations from the average. The reasons for this phenomenon are most likely due to differences at the molecular level, not only between species but also between ureaplasma serovar types. The human two species of *Ureaplasma* spp. contain at least 14 antigenically distinct serovars: *U. parvum* serovars 1, 3, 6, and 14 and *U. urealyticum* serovars 2, 4, 5, and 7 to 13. Serovar types classified as one species differ from each other to varying degrees [28]. The comparative genome study by Paranalov et al. revealed the presence of genes that can support horizontal gene transfer, suggesting that these *Ureaplasma* spp. possess active recombination mechanisms. Therefore, it is possible that ureaplasmas do not exist as stable serovars in their host, but rather as a dynamic population [29]. It is also worth noting that the genes that code for IgA, protease, urease, and phospholipases A and C, which are potential targets for antimicrobial agents, are not present in all the *Ureaplasma* spp. serovars. For example, the genes that code IgA protease and phospholipase A and C have not been identified in the *U. parvum* serovar 3 genome [26,27]. 

It is possible that these features of *Ureaplasma* spp. are the cause of the large deviations from the average observed in several cases in our studies checking the antimicrobial activity of DDC against these bacteria. However, without knowledge of the mechanism of antimicrobial action of DDC, the main metabolite of DSF, against the bacteria *U. urealyticum* and *U. parvum*, we will not solve this problem.

To sum up, the antimicrobial activity of DSF and its main metabolite DDC has been well established in published papers, which concluded that these compounds had a potent activity against Gram-positive and Gram-negative bacteria. *Ureaplasma* spp., a genus of cell wall-free bacteria, cannot be stained by Gram stain. *Ureaplasma* spp. are the smallest self-replicating organisms. This species, although less common, is considered more pathogenic and is usually more drug-resistant. DSF is a compound that was approved by the US Food and Drug Administration (FDA) in 1951 as a drug to treat alcoholism. It has been widely used in clinics for over 70 years as a well-tolerated drug, inexpensive, and without severe side effects. The main biological metabolite of DSF, namely DDC, is generally believed to be responsible for most of the observed pharmacological effects of DSF. 

This study has demonstrated that DDC has activity against *Ureaplasma* spp. This is the first report of the antimicrobial activity of DDC against *U. urealyticum* and *U. parvum* in in vitro cultures. 

## 4. Materials and Methods

### 4.1. General

*Ureaplasma* strains isolated at the Chair of Microbiology of the Jagiellonian University Medical College in Krakow (Poland) from women with infections of the urogenital tract were tested in this study. The study involved 90 strains of *Ureaplasma* spp., including 36 *U. urealyticum* strains and 54 *U. parvum* strains.

### 4.2. Detection and Identification of Ureaplasma spp.

The vaginal swabs were collected from each woman and placed in BioMerieux transport media. Next, they were subcultured in liquid and solid PPLO media with 1% urea, which was prepared in-house according to the previously described procedure [30] and parallelly in Mycoplasma IST 2 kit BioMerieux medium (R2). Identification of *Ureaplasma* spp. was performed via polymerase chain reaction (PCR) using the previously described two pairs of primers specific for the *U. urealyticum* and *U. parvum* genes [31]. 

Reference strains: *U. urealyticum* (ATCC 27816) and *U. parvum* (ATCC 27815) were used as the positive control in the PCR method. Distilled water served as a negative control. To evaluate the PCR reaction, we used agarose gel electrophoresis. Nucleic acid fragments were separated by their length while moving through an agarose matrix. By adding ethidium bromide, which is an intercalating agent, the amplified products were visualized under ultraviolet light.

### 4.3. Research Method Used

In this study, we used the sodium salt of DDC (Merck Life Science Company, Poland; an affiliate of Merck KGaA, Darmstadt, Germany). The effects of three DDC concentrations, 2 µg/mL, 20 µg/mL and 200 µg/mL, on the growth and survival of *Ureaplasma* spp. were studied. DDC is a compound that is highly soluble in water. An aqueous solution of DDC with a concentration of 200 µg/mL was prepared. The concentrations 2 and 20 µg/mL were performed in the culture medium for ureaplasmas. Each of the experiments performed, as well as the testing of the toxic effect of DDC on the cell culture to determine the highest non-toxic concentration of the tested compound, was repeated three times.

The first stage of the study was to determine the initial titer (control system) of each tested *Ureaplasma* strain. This titer was determined via titration of an 18–24 h culture of the tested strain in growth medium without the addition of DDC. The liquid BioMerieux and PPLO and medium contained phenol red at a final concentration of 0.002%. A solution of phenol red is frequently used in cell biology laboratories as a pH indicator. Its color exhibits a gradual transition from yellow over the pH range 6.8 to 8.2. Above pH 8.2, phenol red dye turns red. The addition of a color indicator to the liquid medium is necessary because the growth of *Ureaplasma* spp. does not cause the broth to become turbid.

The titer of the *Ureaplasma* strains was estimated using the color change unit (CCU) test. The study used the phenomenon of metabolic inhibition. *Ureaplasma* spp. exhibit urease activity. Urea hydrolysis plays an important role in the energy metabolism of *Ureaplasma* strains and is their main source of energy. The growth of *Ureaplasma* spp. is objectively determined by their metabolic activity. Urease-utilizing bacteria cells hydrolyze urea in the medium, releasing ammonia, which leads to an increase in the pH of the medium above 8.2. The formation of ammonia as a product of the hydrolytic decomposition of urea is shown by the following reaction: H_2_N-CO-NH_2_ + H_2_O → 2NH_3_ + CO_2_. The increase in the pH value causes the phenol red color change from yellow to red. When the microorganism is sensitive to the test compound, its metabolism is inhibited and the medium remains yellow. If the tested strain is resistant, as a result of its growth, we observe a change in the color of the medium to red. The color change of the substrate was assessed visually.

Titration was performed in 96-well plates using a series of 10-fold dilutions (10^−1^ to 10^−10^). The titer was read after 24 h of incubation at 37 °C. 

Moreover, 1 CCU/mL was defined as the highest dilution of cells at which a color change was detected. The titer was considered to be the highest dilution that produced a color change, when the test result was recorded at the time when color changes were no longer progressive [32]. 

### 4.4. Bacteriostatic Effect of DDC on Ureaplasma Cultures

The cultures of each *Ureaplasma* strain were titrated by making 10-fold serial dilutions (10^−1^ to 10^−10^) in growth medium with the addition of the tested DDC concentrations. Titration was carried out in 96-well plates. The titers were read after 24 h of incubation at 37 °C and expressed as CCU/mL. The results obtained after treatment with DDC were compared to the results obtained from strains not exposed to the DDC compound (control system). 

### 4.5. Bactericidal Activity of DDC 

The culture of each *Ureaplasma* strain was suspended in medium with the tested concentrations of DDC and was incubated for 24 h at 37 °C. After incubation, the culture was centrifuged (Sigma 113 microcentrifuge; 14,000× *g*/20,000 rpm) and washed twice with saline to remove the DDC, and then titration was performed by making 10-fold serial dilutions (10^−1^ to 10^−10^) with growth medium without the addition of DDC similarly as described above. The obtained values were compared with the control titer for a given strain. It should be noted that although the decrease in the bacterial titer correlates with the decrease in bacterial numbers, it does not clearly indicate the bactericidal properties of DDC. This is why, in order to test whether DDC has such properties, we conducted an experiment to check whether the tested strains cultured in the presence of DDC retained the ability to grow in growth medium without the addition of DDC. The results obtained after treatment with DDC were compared to the results obtained from strains not exposed to the DDC compound (control system).

### 4.6. Statistical Analysis

Statistical calculations were carried out with the IBM^®^ SPSS^®^ Statistic 29 (Armonk, NY: IBM Corp, USA). Differences between the study groups were analyzed using Kruskal–Wallis non-parametric tests with Dunn’s post hoc test and the Chi square test (χ2). Statistical significance was defined as *p* < 0.05 for all the tests. Log titer values are presented as medians, with lower and upper quartiles written in brackets.

## 5. Conclusions

The significance of this report is that it is the first study demonstrating that DDC has antimicrobial activity against *Ureaplasma* spp., a genus of cell wall-free bacteria that are often resistant to many antibiotics making treatment difficult. This study opens a new avenue for the use of DDC as a therapeutic against infections caused by *Ureaplasma* spp.

Further research is needed and the results obtained in the present study should be verified in in vivo models of infection. Research is also needed to better understand the mechanisms through which DDC exerts its antimicrobial effects.

## Figures and Tables

**Figure 1 ijms-25-00040-f001:**
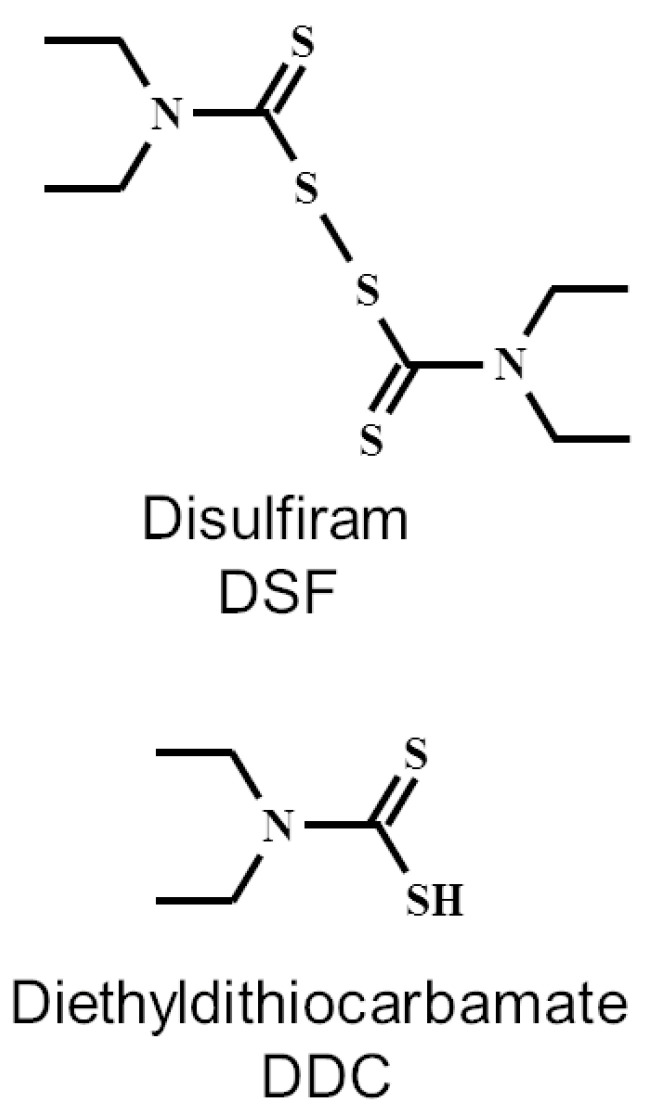
Chemical structures of disulfiram and diethyldithiocarbamate.

**Figure 2 ijms-25-00040-f002:**
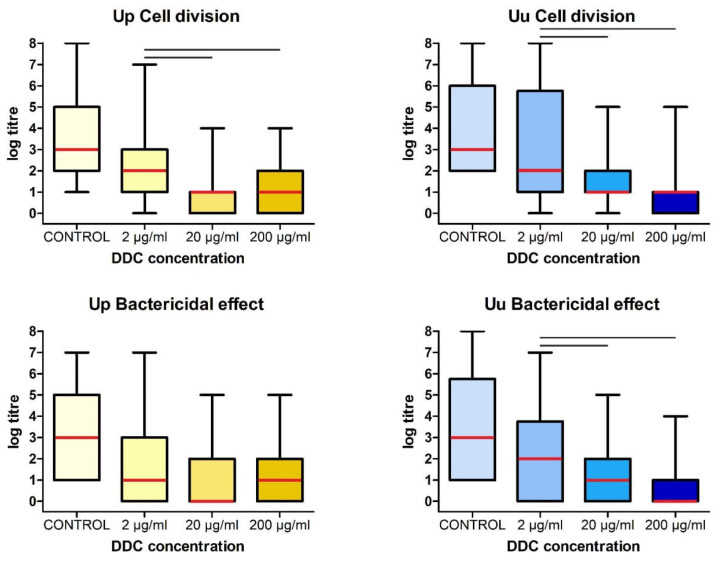
The bacteriostatic and bactericidal effect of DDC on the cells of *Ureaplasma* spp. The graph shows the dependence of the log titer for *U. parvum* (Up) and *U. urealyticum* (Uu) on the DDC dose. Data are shown as the median value (red line) with upper and lower quartiles. The whiskers represent the minimum and maximum values. Groups connected by a line are statistically significant at *p* < 0.05.

**Figure 3 ijms-25-00040-f003:**
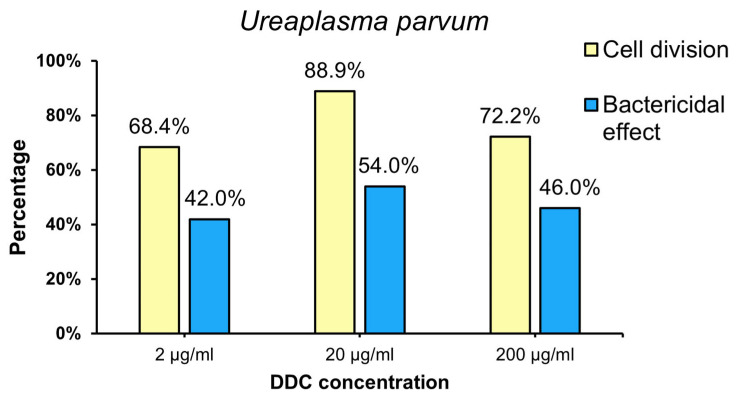
Percentage of *U. parvum* (Up) strain for which the bacteriostatic and bactericidal effects of DDC were demonstrated.

**Figure 4 ijms-25-00040-f004:**
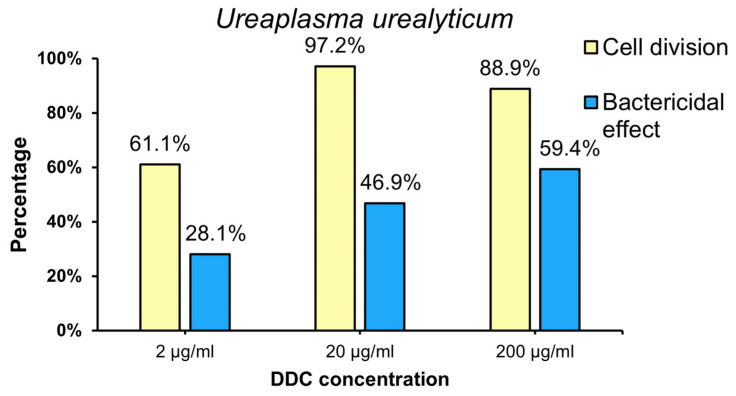
Percentage of *U. urealyticum* (Uu) strain for which the bacteriostatic and bactericidal effects of DDC were demonstrated.

## Data Availability

All data supporting the findings of this study are available within the article and from the corresponding author on reasonable request.

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
