# Peer review of "Antimicrobial Activity of N,N-Diethyldithiocarbamate against Ureaplasma parvum and Ureaplasma urealyticum"

_ijms, 2023, doi:10.3390/ijms25010040_

Round 1

Reviewer 1 Report

Comments and Suggestions for Authors

see attached file

Comments on the Quality of English Language

English is adequate.

Author Response

Reply to Reviewer 1

We are grateful to the Reviewer for valuable comments and thorough review of our paper. Please, find my replies below.

This manuscript describes the effects of the chemical N,N-diethyldithiocarbamate (DDC) on the growth and viability of 36 strains of Ureaplasma urealyticum and 54 strains of Ureaplasma parvum which had been isolated from women with urinary tract infections in Krakow, Poland. ATCC reference strains were used as controls. The identity of strains was confirmed by phenotypic analysis using BioMerieux test strips and by PCR analysis. All of the samples were apparently positive for the enzyme urease, although the specific activity of urease in the various samples was not determined by direct enzyme assays. The data in the paper reflect a preliminary microbiological analysis of the effects of DDC on Ureaplasma spp. While the authors conclude that these effects may be due to the inhibition of urease activity by DDC, they acknowledge at the end of the manuscript that there have been no studies on the inhibition of urease from Ureaplasma by this compound. They also do not cite any studies on the effects of DDC on the ureases of other organisms such as those from jack beans, Proteus mirabilis, Helicobacter pylori, or Staphylococcus saprophyticus which have been used as models for urease inhibition studies. The Introduction and Discussion sections of the paper are relatively long compared to the Materials and Methods section and to the limited data set shown in the Results section.

The Introduction includes a detailed review of the medical effects of Ureaplasma spp. on women. This would be useful to readers who are not familiar with this microorganism and to the possible treatments for its infections. As noted by the authors, these bacteria lack a cell wall and so are not susceptible to beta-lactam or glycopeptide antibiotics. They can be inhibited by fluroquinolones, tetracyclines, chloramphenicol, and macrolides, although antibiotic resistance has been increasing. There is thus considerable interest in new drugs. The introduction ends with a brief introduction to DDC and its precursor disulfiram (DSF) and summarizes their use in the treatment of alcoholism and other diseases. However, the structures of these compounds are not given until the Discussion. Because the focus of this manuscript is on the inhibitory effects of DDC, it would help to move Figure 5 to the Introduction as Figure 1. The source of the DDC used in these experiments is not given in the Materials and Methods and should be included.

     Taking into account the Reviewer's comments, the manuscript has been thoroughly revised. The Introduction has been significantly shortened and re-edited. Extensive fragments regarding the medical effects of Ureaplasma spp. on women have been removed from the Introduction. More space was devoted to the properties of DDC, which is the main biological metabolite of DSF. DSF was approved in 1951 by the US Food and Drug Administration (FDA) as a drug for the treatment of alcoholism. It has been widely used in clinics for over 70 years as a well-tolerated drug, inexpensive, without severe side effects. Repurposing of already approved drugs for new medical uses is therefore a promising alternative strategy as it takes advantage of existing toxicology and pharmacokinetic data from preclinical and clinical trials. Currently, many authors point to DSF as a promising antimicrobial drug.

For this reason, DDC seems to be a compound with great pharmacological potential. As suggested by Reviewer  we moved Figure 5 to the Introduction as Figure 1 and the source of DDC was provided in the Materials and Methods.

     As the Reviewer noted, we have not conducted studies on urease activity. This is just our hypothesis based on the chemical properties of DDC. The problem is that although many authors confirm the antimicrobial activity of DSF and its metabolites, the molecular mechanism of this activity is still not well documented, and in the presented reports the authors mostly provide hypotheses. However, we agree with the Reviewer's suggestion that this should not be done and therefore the fragments regarding our hypothesis were removed from the manuscript.

     Taking into account the Reviewer's comments, the Discussion was also revised and improved, the way of presenting the results was changed, and the information in the Materials and Methods section was presented in a more detailed and clearer way.

The Materials and Methods section describes the methods for determining the titer of the Ureaplasma samples using color changing units (CCU/ml). Since this is different than a standard viable or plate count assay, it would help to explain this method in more detail as it was modified from those given in reference 29. The procedures seem to be similar to those used in a previous study by these same authors (Biernat-Sudolska et al., 2020, reference 43) but needs further explanation. Was the assay based on a change in the color of phenol red due to the formation of ammonium or to the deposition of a red precipitate from the tetrazolium substrate? Were the 96 well plates read by eye or a microtiter plate reader at a certain wavelength? How many replicates of the assays were done for each strain at each DDC concentration? The dilutions in the paper are given as 10–1 to 10–10 with the exponent in a small font rather than a superscript. It would be clearer to use the more standard form of 10–1 to 1010. It appears that for both the bacteriostatic and bactericidal tests, a single incubation time of 24 hours at 37oC was used. The authors should explain why this was done. Why not carry out a longer time course study to look at the effects of 2

     As already mentioned above, the Materials and Methods section has been improved and more detailed information about our experiment has been provided. In fact, a similar procedure was used as in the work on the influence of lipoic acid on ureaplasmas. The recommendations of this procedure were used and the readings were taken after 18-24 hours of incubation. In commercial diagnostic tests, e.g. from BioMerieux, the drug susceptibility results for ureaplasmas are read 24 hours after the strain is inoculated on the medium. Ureaplasma strains tend to autolyze, so prolonged incubation is not used.

The dilutions in the paper are given as 10–1 to 10–10 (the original notation of exponents, subscripts or superscripts often changes the system, e.g. when saving a document saved in Word to a document saved in PDF).

It would help to establish the validity of the assays if the authors were to include a graph at the beginning of the Results section in which they showed the titers of the control strains (ATCC 27816 for U. urealyticum and ATCC 27815 for U. parvum) in the bacteriostatic and bactericidal assays with 0, 2 µg/ml, 20 µg/ml, and 200 µg/ml DDC with the means and standard deviations for as many samples as were tested. This seems particularly important given the wide range of values (the whiskers) shown in Figures 1 and 3 of the Results section. As noted above, a single time point of 24 hours was used and the results of a time course experiment with the control strains would also be helpful.

     Identification of Ureaplasma spp. was done by polymerase chain reaction (PCR) using the previously described two pairs of primers specific for the U. urealyticum and U. parvum genes. Reference strains: U. urealyticum (ATCC 27816) and U. parvum (ATCC 27815) were used only as the positive control in the PCR method. Distilled water served as a negative control. To evaluate the PCR reaction, we used agarose gel electrophoresis. The results obtained after treatment with DDC were compared to the results obtained on strains not exposed to the DDC compound (control system).

The Results section presents the data in a way that is difficult to follow. I think the goal of the authors should be to answer three basic questions about the effects of DDC on the two species of Ureaplasma:

1) is it bacteriostatic or bactericidal?

2) is one species of Ureaplasma more sensitive to DDC than the other?

3) do the effects of DDC show a direct dependence on the concentration used? It is hard to glean the answers to these questions from the way the results are presented.

In the current version of the paper, the method of presenting and describing the results has been improved.

The long whiskers on the top and bottom sides of the box plots in Figures 1 and 3 appear to represent the range of values for each species of Ureaplasma with a certain concentration of DDC. The authors never explain why the range of values is so large. Is this a problem with the assays or does it reflect variations in the sensitivities of the various clinical isolates to DDC? How did the clinical isolates compare to the ATCC type strains? It is not clear whether the data used to generate these box plots included only the strains that showed some inhibition or both those strains that were inhibited and those that were not. The data might look better if the results were restricted to only those strains that showed a bacteriostatic or bactericidal effect. Why are there no lower whiskers in the box plots in Figure 3?

The box plots in the color figures included in the online version of the manuscript are supposed to show the median values for each strain at each concentration with the upper and lower quartiles as red rectangles. However, because of the black lines surrounding the boxes, only the red line in the middle is visible. This appears to be the median value. The median values in Figure 1 seem to be the same for Uu and Up at each concentration: 3 for the controls, 2 for 2 µg/ml, and 1 for 20 µg/ml and 200 µg/ml. This does seem to suggest there is no major difference between the two species and that DDC has a bacteriostatic effect on both. The median values in Figure 3 show more obvious differences between Uu and Up at each concentration. The Up strains appear to be more susceptible to the bactericidal effect of DDC at 2 µg/ml (median value of 1 compared to 2) and at 20 µg/ml (median value of 0 compared to 1). However, the Uu strains were more susceptible at 200 µg/ml (median value of 0 compared to 1).

The Discussion is relatively long and focuses on the general effects of DSF. There is little discussion of the actual data shown in this paper. Why are the variations in the box plots so large? Why might the two species differ in their sensitivities to DDC? How can one explain the large variations for the isolates in their group? While it is plausible that the effects of DDC reflect inhibition of urease activity, there is no direct evidence to support this conclusion. Ureases from other organisms can be inhibited by 2-mercaptoethanol and thiourea, particularly those that contain cysteine at the active site. A more detailed analysis in which the specific activity of the urease in selected isolates was correlated to their sensitivity to a bacteriostatic or bactericidal effect of DDC would be important.

     In line with your suggestions, the Discussion has been thoroughly re-edited and revised. The Discussion devotes more space to the results obtained - as suggested by the reviewer. The thread regarding urease activity has been removed from the discussion. The reviewer is right, this issue requires further research and it was not investigated in our study.

     This paper has been revised according to the Reviewer comments. The Introduction and Discussion have been thoroughly re-edited and revised. The corrected and redacted sections in the Materials and Methods and Results section are highlighted in yellow. Improved way of presenting results (changed Figures). Due to the comments of the Editorial Board, the title of our manuscript has also been changed.

     We hope that the revised version of our manuscript now fulfils the expectations of the Reviewer.

Reviewer 2 Report

Comments and Suggestions for Authors

Dear Editor,

Thank you for your peer review invitation. The manuscript, entitled “In-vitro activity of N,N-diethyldithiocarbamate against Ureaplasma urealyticum and Ureaplasma parvum isolated from women with infections of the urogenital tract” described that all DDC concentrations used showed both bacteriostatic and bactericidal activity against both tested strains. This is the first study showed the bacteriostatic and bactericidal activity of DDC against Ureplasma urealyticum and Ureaplasma parvum. However, due to some drawbacks, my suggestion is rejection.

Comments:

1. In the introduction section, the authors stated that DDC is the main metabolite of disulfiram (DSF) and the use of DSF as a treatment in patients with alcoholism is an unethical therapy because it is connected with the risk of poisoning and even a loss of life. Therefore, I am curious if DDC is poisoning to human? In this study, missing from this study are experiments to verify the toxicity of DDC in humans, which is crucial for drug research.

2. This study only examined the effect of DDC on the inhibition of UU and UP cell differentiation and bactericidal effects. Factors such as different maturity levels of the cells and whether they co-existed with other bacteria were not considered. The experimental design was not perfect.

3. In addition, is DDC more effective than the current medications commonly used in the clinic to treat uu and up? In Figure 4, it can be seen that the bactericidal effect of DDC on UU and UP is only about 50 per cent, an effect that cannot be considered very good.

4. What is the mechanism by which DDC inhibits UU and UP growth differentiation? The authors lack in-depth studies on this aspect as well.

5. The authors can add some data from SEM and watch for changes in the shape of UU and UP.

6. The discussion section does not go far enough, and the placement of Figure 5 in the discussion section is unnecessary.

Comments on the Quality of English Language

English writing could be improved.

Author Response

Reply to Reviewer 2

We are grateful to the Reviewer for valuable comments and thorough review of our paper. Please, find my replies below.

Comments:

  1. In the introduction section, the authors stated that DDC is the main metabolite of disulfiram (DSF) and the use of DSF as a treatment in patients with alcoholism is an unethical therapy because it is connected with the risk of poisoning and even a loss of life. Therefore, I am curious if DDC is poisoning to human? In this study, missing from this study are experiments to verify the toxicity of DDC in humans, which is crucial for drug research.

Disulfiram (Antabuse, DSF) was approved in 1951 by the US Food and Drug Administration (FDA) as a drug for the treatment of alcoholism. It has been widely used in clinics for over 70 years as a well-tolerated drug, inexpensive, without severe side effects.

DSF is used as oral tablets or as an implant. DSF is an inhibitor of aldehyde dehydrogenase (ALDH), which catalyzes the oxidation of alcohol to acetaldehyde. During treatment with DSF, you must not drink alcohol because, as a result of inhibiting ALDH activity, the concentration of acetaldehyde in such a patient may increase up to 10 times compared to the standard value. Some 5 to 10 minutes after alcohol intake, the patient may experience the effect of severe “hangover” that may last from 30 min to several hours. Symptoms may include flushing of the skin, accelerated heart rate, shortness of breath, vomiting, nausea, throbbing headache, mental confusion and circulatory collapse.

This therapy is called aversion therapy because the patient maintains abstinence for fear of feeling bad. That is why some doctors and scientists call these therapies unethical, because the patient does not drink for fear of feeling very bad. So if you don't drink alcohol, DSF is a safe drug.

The main biological metabolite of DSF is N,N-diethyldithiocarbamate (DDC). DSF is rapidly and completely converted to two molecules of DDC by serum albumin and erythrocyte enzymes. Thus, DDC, as the main in vivo metabolite of DSF, is also safe for humans.

Moreover, DDC is generally believed to be responsible for most of the observed pharmacological effects of DSF. Recently, it has been shown that DSF and its metabolites can exert antimicrobial effects.

Initially, the this study we determined DDC concentrations that were non-toxic for cell cultures and only these doses were used in further experiments.

Taking into account the Reviewer's comments, the manuscript has been thoroughly revised. The Introduction has been significantly shortened and re-edited. We have provided information regarding DSF and its metabolite DDC more precisely and most clearly.

  1. This study only examined the effect of DDC on the inhibition of UU and UP cell differentiation and bactericidal effects. Factors such as different maturity levels of the cells and whether they co-existed with other bacteria were not considered. The experimental design was not perfect.

We agree with the Reviewer's comment that every test determining drug sensitivity must refer to the so-called pure breeding of a given species. The culture cannot be contaminated with other bacteria. In our research, we also used pure breeding.

We also understand the validity of the Reviewer's comment that all cultures of the tested strains should be at a similar stage of development. However, due to the fact that ureaplasmas grow without turbidity in liquid media, the principle of equal density could not be applied as in the classic drug susceptibility testing (ureplasmas are atypical bacteria). However, in order to minimize the test error, in our experiment we always used an 18-hour culture with a known titer. This type of procedure is used by many authors who study Urepalasma spp.

  1. In addition, is DDC more effective than the current medications commonly used in the clinic to treat uu and up? In Figure 4, it can be seen that the bactericidal effect of DDC on UU and UP is only about 50 per cent, an effect that cannot be considered very good.

Our experiments show that DDC has both bactericidal and bacteriostatic effects.

We agree with the Reviewer's comment that the bactericidal effect of DDC on Uu and Up is only about 50 per cent and an effect that cannot be considered very good.

On the other hand, the analysis of the obtained results clearly showed that the bacteriostatic effect of DDC on U. urealyticum and U. parvum strains is much stronger than the bactericidal effect. DDC at a dose of 20 μg/ml is bacteriostatic for 97.2% of U. urealyticum cells and 88.9% of U. parvum cells. Therefore, the obtained results are promising.

  1. What is the mechanism by which DDC inhibits UU and UP growth differentiation? The authors lack in-depth studies on this aspect as well.

The antimicrobial activity of DSF and its main metabolite DDC has been well established in published papers which concluded that these compounds had a potent activity against Gram-positive and Gram-negative bacteria. Nevertheless, the mechanism of the antimicrobial activity of DSF and DDC against these bacteria is not well understood. However, there are no studies on the antimicrobial activity of DDC against Ureaplasma spp. so far. The presented paper is the first report of the antimicrobial activity of DDC against U. urealyticum and U. parvum in in vitro cultures. Recognizing the mechanism of its action will be difficult, considering that despite a relatively large number of studies indicating antimicrobial activity against Gram-positive and Gram-negative bacteria, this mechanism is still not well researched. Ureplasmas are atypical bacteria. They are the smallest self-replicating organisms, and due to the lack of a cell wall they cannot be stained by Gram stain. We briefly discuss this issue.

The significance of our report is that it is the first study demonstrating that DDC has antimicrobial activity against Ureaplasma spp., a genus of cell wall-free bacteria which are often resistant to many antibiotics making treatment difficult. Further research is needed to better understand the mechanisms through which DDC exerts its antimicrobial effects against these bacteria. We intend to conduct these studies.

  1. The authors can add some data from SEM and watch for changes in the shape of UU and UP.

We agree with the comments regarding the possibility of using electron microscopy. Ureaplasmas are uniquely small in size. Their cells are not visible in an optical microscope. Due to the lack of cell wall structures, we cannot stain them, which would facilitate the observation of cell morphology. Certainly, and we agree with the reviewer's comment, it would be interesting to use other optical devices. Unfortunately, we were unable to conduct research using an electron or scanning microscope. Electron microscopes are very powerful tools for visualizing biological samples, including very small organisms. However, they cannot be alive. Biological samples must undergo complex preparation steps, and the preparation process kills tissue and may also cause changes in the appearance of the sample. Therefore, a big challenge is to develop a method of preparing a biological sample for visualization in an electron microscope so that its morphology remains preserved.

  1. The discussion section does not go far enough, and the placement of Figure 5 in the discussion section is unnecessary.

Thank you for pointing out that the Discussion needs refinement. The authors of the paper decided to expand and reword it. Thank you for pointing out that the Discussion needs refinement. The Discussion was thoroughly revised and improved. As suggested by Reviewer 1, we moved Figure 5 to the Introduction as Figure 1.

This paper has been revised according to the Reviewer comments. The Introduction and Discussion have been thoroughly re-edited and revised. The corrected and redacted sections in the Materials and Methods and Results section are highlighted in yellow. Improved way of presenting results (changed Figures). Due to the comments of the Editorial Board, the title of our manuscript has also been changed.

We hope that the revised version of our manuscript now fulfils the expectations of the Reviewer.

Round 2

Reviewer 1 Report

Comments and Suggestions for Authors

This revision is an improvement over the original manuscript.  It would benefit from careful copy-editing by a native English speaker.  There are places in the headings to sections 3.1.1 and 3.1.2 and to 3.2.1 and 3.2.2 where the name of the genus or the names of the two species are not given in italics.

Was the DDC dissolved in water, buffer, or medium?  The authors still have not indicated how many replicates were run for each strain under each test condition.  Figure 2 is clearer but in the text, the authors give the results as the median log titer followed by the quartile values.  For example, 3 (2, 5) for the Up control culture in the bacteriostatic assay.  They never state that the numbers in parentheses are the upper and lower quartile values, although one can figure this out from the figure.

I think there is an error in section 3.2.2 where the authors state there was no significant difference in the bacteriocidal effect of DDC against U. urealyticum between 200 ug/ml and 2 ug/ml (59.4% compared to 28.1%). 

While the authors discuss the fact that some of the results were not statistically significant, they never directly address the question of why the range of values in in Figure 2 is so large.  It would help to show that this is not due to variations in the metabolic assay itself. 

Section 3.3 is a summary of the results from the previous two sections.  I think that it might better be used as the first paragraph of the Discussion. 

Krakow is still spelled with both a K and a C.

Comments on the Quality of English Language

The quality of the English language is adequate but the text would benefit from copy editing.

Author Response

Reply to Reviewer 1

We are grateful to the Reviewer for valuable comments and thorough review of our paper.

Please, find my replies below.

There are places in the headings to sections 3.1.1 and 3.1.2 and to 3.2.1 and 3.2.2 where the name of the genus or the names of the two species are not given in italics.

It has been corrected.

Was the DDC dissolved in water, buffer, or medium? The authors still have not indicated how many replicates were run for each strain under each test condition.

We thank the Reviewer for pointing out the missing information. DDC is a compound that is highly soluble in water. An aqueous solution of DDC with a concentration of 200 µg/ml was prepared. The concentrations 2 and 20 µg/ml, were performed in the culture medium for ureaplasmas. Each of the experiments performed, as well as testing the toxic effect of DDC on cell culture to determine the highest non-toxic concentration of the tested compound, was repeated three times. This information has been entered in the Materials and Methods section.

Figure 2 is clearer but in the text, the authors give the results as the median log titer followed by the quartile values.  For example, 3 (2, 5) for the Up control culture in the bacteriostatic assay. They never state that the numbers in parentheses are the upper and lower quartile values, although one can figure this out from the figure.

After the first review, the following sentence was added to Section 2.6 at the very end "Log titre values are presented as medians with lower and upper quartiles”. In the current version of the manuscript, we have added the sentence "Log titre values are presented as medians with lower and upper quartiles quartiles written in brackets" at the end of Section 2.6.

I think there is an error in section 3.2.2 where the authors state there was no significant difference in the bactericidal effect of DDC against U. urealyticum between 200 ug/ml and 2 ug/ml (59.4% compared to 28.1%).

Thank you for this comment. Thanks to this Reviewer's comment, we carefully reviewed the Results section again and made some corrections to the text. We hope that this has made the presentation of the results better and clearer for readers.

While the authors discuss the fact that some of the results were not statistically significant, they never directly address the question of why the range of values in in Figure 2 is so large. It would help to show that this is not due to variations in the metabolic assay itself. 

We have added an additional paragraph to the Discussion to address this issue.

Section 3.3 is a summary of the results from the previous two sections. I think that it might better be used as the first paragraph of the Discussion.

As suggested by the Reviewer, the Section 3.3. was moved to the Discussion as the first paragraph

Krakow is still spelled with both a K and a C.

It has been corrected.

All changes are highlighted in green.

Reviewer 2 Report

Comments and Suggestions for Authors

The authors had made revision accordingly.

Comments on the Quality of English Language

The English writing meets the standard.

Author Response

Reply to Reviewer 2

We are grateful to the Reviewer for all valuable comments and thorough review of our paper.
